# ADAM12-Generated Basigin Ectodomain Binds β1 Integrin and Enhances the Expression of Cancer-Related Extracellular Matrix Proteins

**DOI:** 10.3390/ijms25115871

**Published:** 2024-05-28

**Authors:** Kasper J. Mygind, Denise Nikodemus, Sebastian Gnosa, Ramya Kweder, Nicolai J. Wewer Albrechtsen, Marie Kveiborg, Janine T. Erler, Reidar Albrechtsen

**Affiliations:** 1Biotech Research and Innovation Centre (BRIC), Faculty of Health and Medical Sciences, University of Copenhagen, 2200 Copenhagen, Denmark; kmygind@sund.ku.dk (K.J.M.); denise.nikodemus@biologie.uni-freiburg.de (D.N.); seg@minervaimaging.com (S.G.); ramya.kweder@sund.ku.dk (R.K.); marie.kveiborg@sund.ku.dk (M.K.); janine.erler@bric.ku.dk (J.T.E.); 2Department of Clinical Biochemistry, Copenhagen University Hospital-Bispebjerg, 2400 Copenhagen, Denmark; nicolai.albrechtsen@regionh.dk

**Keywords:** CD147/Basigin, disintegrin and metalloproteinase, extracellular matrix

## Abstract

Desmoplasia is a common feature of aggressive cancers, driven by a complex interplay of protein production and degradation. Basigin is a type 1 integral membrane receptor secreted in exosomes or released by ectodomain shedding from the cell surface. Given that soluble basigin is increased in the circulation of patients with a poor cancer prognosis, we explored the putative role of the ADAM12-generated basigin ectodomain in cancer progression. We show that recombinant basigin ectodomain binds β1 integrin and stimulates gelatin degradation and the migration of cancer cells in a matrix metalloproteinase (MMP)- and β1-integrin-dependent manner. Subsequent in vitro and in vivo experiments demonstrated the altered expression of extracellular matrix proteins, including fibronectin and collagen type 5. Thus, we found increased deposits of collagen type 5 in the stroma of nude mice tumors of the human tumor cell line MCF7 expressing ADAM12—mimicking the desmoplastic response seen in human cancer. Our findings indicate a feedback loop between ADAM12 expression, basigin shedding, TGFβ signaling, and extracellular matrix (ECM) remodeling, which could be a mechanism by which ADAM12-generated basigin ectodomain contributes to the regulation of desmoplasia, a key feature in human cancer progression.

## 1. Introduction

The extracellular matrix metalloproteinase inducer (EMMPRIN), also termed CD147/basigin, is a glycoprotein belonging to the immunoglobulin family [1,2]. Four variants (basigin 1–4) exist, but the most abundant and well-studied is isoform 2 (henceforth termed basigin). Basigin (BSG) is a 269 amino acid long type I transmembrane protein composed of an N-terminal signal sequence, a 186 residue-long extracellular portion consisting of two Ig-like domains (D1 and D2), a transmembrane domain, and a C-terminal cytoplasmic domain [3]. It has been shown to exert several different functions at the plasma membrane, acting as a homo- or heterodimer in either cis- or trans-configurations [3,4,5,6,7,8].

Apart from numerous recent papers on its role in SARS-CoV-2 infection [9], most studies on BSG focused on its ability to activate matrix metalloproteases (MMPs) and, in this way, stimulate cancer cell invasion [3]. Of note, BSG, expressed by tumor cells, induces the expression of MMPs in fibroblasts or other non-malignant stromal cells through direct or indirect cellular interactions [2,3,10]. Moreover, BSG stimulates cancer cell invasion through mechanisms involving the interaction with cell-surface β1 integrin [11,12]. Finally, BSG promotes tumor progression by facilitating the translocation of monocarboxylate transporters to the plasma membrane, thereby increasing the aerobic glycolysis of tumor cells [13].

Recently, increased attention has been paid to a soluble form of basigin (sBSG). Particularly, sBSG has been detected in the blood of cancer patients and clinical data show a correlation between increased levels of sBSG and poor patient prognosis [14,15]. sBSG exists as a full-length form released in exosomes [16], or as a proteolytic fragment shed from the membrane by the MMP14-mediated cleavage between the D1 and D2 domains [17]. In addition, we previously showed that the shedding of BSG by ADAM12 generates a larger soluble fragment, comprising the major part of the ectodomain of the molecule and, hence, both the D1 and D2 domains [18].

ADAM12 plays an important role in cancer. It is frequently upregulated in malignant tumors, with increased levels of expression associated with a poor prognosis [19,20,21]. In addition to BSG, transmembrane ADAM12 is capable of shedding a number of other transmembrane molecules, such as some epidermal growth factor receptor (EGFR) ligands [19,22] and adhesion molecules like VE-cadherin [23]. Moreover, ADAM12 exerts pro-tumorigenic effects independent of its protease function, partly by regulating integrin and MMP14 activity [24,25].

We recently demonstrated the presence of sBSG in blood from patients with bladder cancer [18], which expresses high levels of ADAM12 [26]. Moreover, a significant correlation between the levels of ADAM12 and sBSG in serum from prostate cancer patients has been reported [14]. To further explore the putative biological role of ADAM12-generated sBSG, we produced recombinant sBSG (rsBSG) corresponding to the ADAM12-generated ectodomain, and tested its effect on cell migration. We were able to show that rsBSG had a stimulating effect on TGFβ-responsive proteins such as MXRA5, and several ECM proteins both in vitro and in vivo, mimicking the desmoplastistic changes seen in human cancer. Importantly, changes in the tumor stroma have long been associated with a poor clinical outcome in patients with cancer [27].

## 2. Results

### 2.1. ADAM12-Generated Soluble BSG (sBSG) Stimulates In Vitro Cancer Cell Migration and Invasion

Purified recombinant soluble BSG (rsBSG), corresponding to the cleaved fragment generated by ADAM12 and the shorter BSG fragment only containing the membrane-proximal D1 domain (rsD1), corresponding to the MMP-14-cleaved fragment are shown in Figure 1A. Treating HT1080 cells with rsBSG increased cell migration in an in vitro wound closure assay, as compared to control-treated cells. In contrast, no statistically significant effect of rsD1 on cell migration was observed (Figure 1B,C). To test the effect of rsBSG on cell invasion, we used a 3D spheroid invasion assay in which cellular outgrowth is measured [28]. Like for cell migration, we observed that treatment with rsBSG, but not rsD1, led to an increased invasion (Figure 1D,E). To investigate if the cellular outgrowth was similar to invasion through basement membrane, we let cells treated with control or rsBSG invade through Matrigel. We observed a significant increase in invasive capacity in rsBSG-treated cells (Figure 1F). We next asked whether ADAM12 is capable of inducing migration through soluble BSG (sBSG). Conditioned media from cells transfected with ADAM12 were cleared of sBSG using an N-terminal-specific BSG antibody and the reduction in sBSG was confirmed by a western blot analysis (Figure 1G). We observed that HT1080 cells treated with media devoid of sBSG showed decreased cell migration compared to control IgG cleared media (Figure 1H,I).

### 2.2. rsBSG Enhances Integrin Signaling Resulting in Increased Cell Migration and Gelatin Degradation

BSG is known to interact with several cell-surface proteins, in particular, integrins [11]. The binding between BSG and β1 integrin involves the membrane-proximal D2 domain (specifically amino acid Asp179) of BSG and the metal ion-dependent adhesion site (MIDAS) of β1 integrin, which is also where proteins with an RGD motif bind [29]. To test whether sBSG binds to cell-surface β1 integrin, we added rsBSG to HT1080 WT cells and performed co-immunoprecipitation using anti-BSG and anti-β1 integrin antibody in cell lysates from HT1080 WT cells (Figure 2A,B). We observed that, indeed, β1 integrin can bind rsBSG. Because rsBSG is able to bind β1 integrin, we investigated if rsBSG induces integrin-mediated downstream signaling. HT1080 cells were starved, followed by stimulation with BSA (control) or rsBSG for 30 min. As demonstrated in Figure 2C,D, rsBSG stimulated Focal Adhesion Kinase (FAK) activation (p397FAK), known to be critically involved in the signaling pathway during cell adhesion and migration [30]. Since active focal adhesions (FAs) are known to be rich in phosphorylated tyrosine (pTyr), we treated cells with rsBSG for 60 min and stained for pTyr. When examining the FA complexes, we observed that cells treated with rsBSG showed slightly larger FAs (Figure 2E), indicative of increased activity. The ability of cells to degrade the extracellular matrix is often required for efficient cell migration. We tested whether rsBSG was able to increase ECM degradation. Indeed, HT1080 cells seeded on gelatin showed increased degradation when stimulated with rsBSG (Figure 2F,G). Since gelatin degradation is known to be dependent on MMPs [25], we tested the effect of the MMP inhibitor (GM6001). As expected, we observed a clear decrease in rsBSG-induced gelatin degradation in the presence of GM6001 (Figure 2F,G). In line with the observed activation of β1 integrin by rsBSG, the β1-integrin-inhibitory antibody AIIB2 also strongly inhibited gelatin degradation (Figure 2F,G). For comparison, we tested whether the rsD1 fragment was able to induce gelatin degradation, but found no significant changes compared to the BSA control (Figure 2H,I).

### 2.3. rsBSG Stimulates ADAM12 Expression through TGFβ Signaling

It is known that TGFβ1 signaling is a major inducer of ADAM12 expression [23,31]. Further, as a positive feedback loop between BSG and TGFβ has been reported [32], we hypothesized that rsBSG might influence the expression of ADAM12 through TGFβ stimulation. As shown by qRT-PCR, ADAM12 mRNA expression was indeed increased approximately three-fold in HT1080 cells treated with rsBSG, as compared to cells treated with BSA or rsD1 (Figure 3A). We next tested the implication of TGFβ signaling on rsBSG-induced ADAM12 expression and found that it was completely abolished by the inhibition of TGFβ signaling with the TGFβ-RI Kinase Inhibitor VI SB4311542 (Figure 3B). Moreover, the increase in ADAM12 mRNA expression upon rsBSG treatment was significantly inhibited when cells were stimulated with rsBSG in combination with metalloproteinase inhibitor GM6001 (Figure 3C). Interestingly, the mRNA expression level of TGFβ3 was significantly increased when HT1080 cells were treated with rsBSG (Figure 3D), whereas rsBSG had no effects on TGFβ1 mRNA expression (Figure 3E).

### 2.4. rsBSG Stimulates Fibronectin Deposition in HT1080 Cells

Interestingly, following stimulation of HT1080 WT cells with rsBSG or the BSA control for 24 h, we observed a significant increase in the formation of filamentous fibronectin in rsBSG-treated HT1080 WT cells, as demonstrated by immunostaining (Figure 4A). A western blot analysis showed a moderate but significant increase in fibronectin protein levels when cells were treated with rsBSG (Figure 4B,C), and the expression of fibronectin mRNA were also found to have increased upon rsBSG treatment (Figure 4D).

### 2.5. Increased Deposits of Collagen Type 5 in the Stroma of MCF7 Tumor in Mice Is ADAM12-Dependent

It is well-established that human breast carcinomas exhibit increased deposits of extracellular matrix (ECM) proteins such as fibronectin and different collagens in the fibrous stroma, a condition termed desmoplasia [33]. Previously, we demonstrated that ADAM12 sheds the extracellular part of BSG [18]. Based on the previous findings, this prompted us to study the effect of ADAM12 on ECM molecules in breast carcinomas, using an experimental mouse model. Specifically, we used a previously characterized MCF-7 tet-off model system [34]. First, MCF-7 cells were cultured for 3–4 days with or without the tetracycline-derivative doxycycline (Dox). As demonstrated in Figure 5A, ADAM12 expression was reduced when MCF-7 cells were grown in the presence of Dox, as compared to cells grown without Dox. A clear decrease in MXRA5 mRNA expression (a marker of TGFβ signaling) was observed in the Dox-treated cells (Figure 5B). Interestingly, mRNA expression of collagen type 5a2 was reduced when the cells were grown in the presence of Dox (Figure 5C). For the subsequent in vivo experiments, the MCF7 cells were injected into the mammary gland of nude mice. One group of mice received Dox daily in the drinking water while the control group did not. The overexpression of ADAM12 in MCF7 cells resulted in a significantly higher tumor burden compared with the control MCF7-A12 delta cyt + Dox mice after 2–3 weeks [25]. The tumor tissue was excised, fixed in formalin, embedded in paraffin, and processed for immunostaining for collagen type 5. As demonstrated in Figure 5D, we found that mouse tumor tissue without Dox (expressing ADAM12) exhibited fibrillary collagen type 5 deposits, whereas no collagen type 5 was found in tumor tissue from mice which had received Dox in the drinking water and thus did not express ADAM12 (Figure 5D).

Finally, we examined the expression pattern of ADAM12 and collagen type 5 in cDNA arrays from human cases of breast carcinoma. More specifically, we examined the expression profile in datasets from four different cohorts, as described in Materials and Methods. Interestingly, a positive correlation was found between the mRNA expression of ADAM12 and the expression of COL5A1, COL5A2, and COL5A3, as well as the matrix deposition regulator MXRA5. The gene expression profiles were assessed by a simple linear regression analysis and a positive Pearson correlation shown between ADAM12 and COL5A2 (E, F, G, and H), between ADAM12 and COL5A1 (I), and between ADAM12 and COL5A3 (J) and MXRA5.

## 3. Discussion

Cancer progression involves several critical steps, including tumor cell invasion and migration, as well as profound changes in the surrounding stroma called desmoplasia [27,35]. It is well-documented that both ADAM12 [20] and BSG [3] promote cancer progression, at least in part by regulating integrin and MMP activity. The findings reported here add new insight into the underlying molecular mechanisms, indicating that sBSG, proteolytically shed by ADAM12 from the cancer cell surface [18], binds β1 integrin at the same or neighboring cells to induce MMP activation and β1 integrin-dependent cancer cell migration. Interestingly, we observed that purified rsBSG, in addition to inducing β1 integrin-driven cancer cell migration, also stimulates TGFβ3 expression, leading to the increased expression and deposits of matrix proteins fibronectin and collagen type 5, mimicking the desmoplastic response in human cancer [33].

The desmoplastic response in human cancer is characterized by the growth of dense connective tissue [27,36] and, importantly, has been correlated to a poor prognosis for patients [29]. Many studies have been performed to understand its formation as well as its role(s) in tumor progression [36,37] and putative barrier function during treatment [29]. It has previously been shown that a TGFβ-induced upregulation of collagen type 5 in the stroma of breast carcinoma leads to an increased deposition of collagen type 1. Here, we provide evidence that, in addition to already known factors such as PDGF [37] and LOX [36], ADAM12 and rsBSG modulate the extracellular matrix as evidenced by the increased deposits of type 5 collagen in the stroma around the MCF7 tumor, producing increased ADAM12 in a nude mice tumor model system (Figure 5).

Positive feedback loops between both BSG and TGFβ [34], as well as between ADAM12 and TGFβ signaling [33], have been independently reported. Since the pathway involves TGFβ signaling, our data further suggest a novel positive feedback loop whereby sBSG stimulates TGFβ signaling and the subsequent ADAM12 expression to sustain its own shedding. Interestingly, the effect of rsBSG on ADAM12 expression relies on both TGFβ signaling and MMP activity shown in Figure 3. The involvement of MMP activity for the signaling could be due to a release of TGFβ from its complex, which is also known to be bound through RGD binding to the integrin [38,39,40,41].

It has previously been demonstrated that ADAM12 augments both MMP-14- and MMP-2 activation [25], resulting in the release of TGFβ from β1 integrin [42]. Given that rD1 had no effect on TGFβ1 or TGFβ3 expression, the induced ADAM12 expression appears to involve both functional entities of the BSG ectodomain—i.e., the activation of MMPs by the distal part of the molecule and TGFβ1 signaling, which presumably is induced by the observed increase in TGFβ3 expression that requires the entire ectodomain. Yet, the molecular mechanism underlying the increased TGFβ3 expression remains to be revealed.

Several homophilic or heterophilic interactions between BSG molecules or between BSG and other transmembrane proteins, respectively, have been implicated in the pleiotropic effects of BSG [3,6,43]. Of note, the ability to induce MMP activity has been attributed to the first extracellular Ig domain (D1) binding to full-length transmembrane BSG [17], whereas the extracellular-membrane-proximal D2 domain was shown to bind the MIDAS motif of **β**1 integrin [29]. Interestingly, MMP14 only sheds the D1 domain, but not the integrin-binding D2 domain [17], whereas the ADAM12-generated sBSG contains both domains [18]. In line with these findings, we show that rsBSG, corresponding to the ADAM12-generated ectodomain, co-immunoprecipitates with **β**1 integrin and the binding of rsBSG to the cell surface. Supporting the fact that ADAM12-generated sBSG induces integrin activation as shown in Figure 2 by focal adhesion activation, we found that rsBSG stimulates cell migration, whereas a recombinant fragment corresponding to MMP14-generated sBSG had no effect. Given that cancer invasion relies on both cancer cell migration and extracellular matrix degrading MMPs, which are predominantly produced by non-cancerous stromal cells [44], we speculate that the major role of the MMP14-generated, short sBSG variant is to induce MMP activity in surrounding stromal cells, whereas the ADAM12-generated, long sBSG variant [18] mainly activates the migration of cancer cells in an auto- or paracrine manner. This would be in line with studies indicating that the MMP14-generated sBSG fragment, released by cancer cells, acts to induce MMP2 activity in neighboring fibroblasts [17,45].

Finally, we examined human breast cancer data of cDNA arrays and found a strong correlation between ADAM12 mRNA expression and the three chains of collagen type 5. Thus, in keeping with the experimental data presented here, we suggest that ADAM12 is involved in the processing and formation of desmoplasia through its stimulation of type 5 collagen synthesis important for the augmentation of the type 1 collagen present in the stroma around tumors [27].

In conclusion, the findings reported here add to the complexity of mechanisms whereby BSG exerts its cancer-related functions. In addition to acting through both homophilic and several different heterophilic interactions, our data suggest that the differential cleavage of the BSG ectodomain elicits distinct functions during tumor progression and point to an important role of ADAM12-generated sBSG in both cancer cell migration and the desmoplastic response in human cancer.

## 4. Material and Methods

### 4.1. Antibodies and Reagents

Goat polyclonal antibody N-19 (sc-9752), rabbit polyclonal antibody H-200 (sc-13976), mouse anti-collagen type 5a1 (SC-166155), and mouse monoclonal (sc-21746) against BSG were from Santa Cruz Biotechnology (Dallas, TX, USA). The rabbit polyclonal antibody PA5-29787 against BSG was from Thermo Fischer Scientific (Waltham, MA, USA). Mouse monoclonal anti-Strep tag (SAB2702216) and rat monoclonal anti-β1 integrin (AIIBII) antibodies were from Sigma-Aldrich (St. Louis, MO, USA), mouse monoclonal antibody against actin (MAB1501) were from Millipore Chemicon (Burlington, MA, USA), rabbit anti-p397 FAK (44-625G) from Invitrogen (Waltham, MA, USA), mouse anti-phospho Tyrosine (clone 4G10) (16-452) and Rabbit anti-fibronectin (MAB2033) from Millipore (16-452) (Burlington, MA, USA), mouse anti-FAK (610088) from BD Bioscience (Franklin Lakes, NJ, USA). Secondary antibodies used were horseradish peroxidase (HRP)-conjugated goat anti-mouse, goat anti-rabbit, and rabbit anti-goat immunoglobulins from DAKO A/S (Glostrup, Denmark). Alexa Fluor 488-conjugated goat anti-Rabbit IgG and Donkey anti-mouse IgG were from Invitrogen (Taastrup, Denmark). The TGF-βRI Kinase Inhibitor VI (SB431542) was from Sigma-Aldrich (St. Louis, MO, USA). The metalloprotease inhibitor GM6001 and all other chemicals were from Merck KGaA (Darmstadt, Germany).

### 4.2. Plasmids

Mammalian expression cDNA constructs encoding human ADAM12 delta cyt were previously described [25]. Extracellular fragments of human BSG, lacking the transmembrane and intracellular domains (GenBank: BAA08109.1, aa22-207 rsBSG, and aa22-95 r D1) were cloned into a modified Sleeping Beauty transposon vector V358 [46] with an N-terminal double Strep-tag II.

### 4.3. Cell Culture

HEK293, HT1080, HeLa, MDA-MB-231, and MCF7 cell lines were obtained from ATCC and cultured as previously described [23]. HEK293 cells stably expressing the vitronectin receptor αVβ3 integrin (called 293-VnR) were previously described [47].

### 4.4. Recombinant Expression and Purification of Basigin Fragments

rsBSG and rD1 were produced in HEK293 cells, stably transfected using the Fugene HD Transfection Reagent (Promega, Madison, WI, USA) and selected with 4 μg/mL puromycin (Sigma) for 3 days. Protein expression was induced using 0.5 μg/mL doxycycline. Secreted BSG fragments were purified by the IBA Life Sciences Strep-Tactin Sepharose^TM^ (Resin) and dialyzed against 1xPBS.

### 4.5. Immunofluorescence Staining

Visualization of BSG by immunofluorescence staining was performed as previously described [25]. For staining of adherent cells, fixation by either cold methanol or 4% paraformaldehyde was used. Fluorescence imaging was performed using an inverted Zeiss Axiovert 200 Apotome system, equipped with a 63/1.4 Plan-Apochromat water immersion objective, or Zeiss LMS800 confocal laser scanning microscope equipped with 40/1.4 Plan-Apochromat oil immersion objective. Images were processed using the Axiovision program (Carl Zeiss, Oberkochen, Germany) and MetaMorph software (MM45, Molecular Devices, San Jose, CA, USA).

### 4.6. Wound Healing and 3D Spheroid Invasion Assays

HT1080 cells were seeded in 12-well cell culture plates (Corning) at a concentration of 2 × 10^4^ cells/well. When cells reached confluence, sterile pipette tips were used to make scratch wounds across each well. The wells were washed twice with PBS, and serum-free medium (SFM) with 0.1 µg/mL rsBSG, 0.2 µg/mL rD1, or 0.2 µg/mL BSA was added. Images of 4 fields of view per scratch wound were taken at 0 and 6 h, and cell migration was quantitated using ImageJ software 3D spheroid invasion assay as previously described [48]. MDA-MB 231 cells were seeded in hanging drops using 0.24 mg/mL MethoCel A4M (Dow Chemical, MI, USA) and 28.2 μg/mL bovine type I collagen (Corning, NY, USA). After 5 days, single spheres were embedded in a 20 μL drop containing 1.5 mg/mL bovine type I collagen and 5× collagen buffer (5× MEM powder, 1M HEPES pH 7.5, 2% NaHCO_3_). After 1 h, as the collagen drops polymerized, serum-free Dulbecco’s Modified Eagle’s Medium (DMEM) (Invitrogen) with and without 10 ug/mL rsBSG or rsD1 were added on top. Spheroid invasion was assessed 24 h later using brightfield images at 10× magnification. ImageJ software V1.54F (FIJI) was used to measure the cancer cell invasion area.

### 4.7. Transwell Invasion Assay

Invasion chambers with 8 μm pore size precoated with Matrigel were used (Corning BioCoat Matrigel #354480). HT1080 cells were seeded in the insert in serum-free media. The inserts were transferred into a 24-well plate containing 1 ml serum-free media with 0.2 μg/mL BSA or rsD2 BSG and incubated for 16–24 h at 37 °C. After incubation, invaded cells were fixed in 4% paraformaldehyde, washed in PBS, and stained with 1% crystal violet. Invasion was determined by counting cells in 10 pictures from each transwell insert (Axioplan 2, Zeiss).

### 4.8. In Situ Gelatinase Assay

In situ gelatinase assay was performed as previously described [25]. In brief, HT1080 cells were seeded in 3.5 cm dishes (Nunc) coated with gelatin (13.7 μg/mL) coupled to Oregon green 488 dye (G-13186), from Molecular Probes (Life Technologies, Taastrup, Denmark). Two hours later, gelatin degradation was quantified by measuring the area of black holes in the fluorescent gelatin relative to the total area, using ImageJ.

### 4.9. Immunoprecipitation and Western Blot Analysis

Protein immunoprecipitation and western blot analysis were performed as previously described [49]. In brief, HT1080 cells were treated with rsBSG for 6 h, washed in PBS, and proteins extracted in RIPA buffer (20 mM Tris-HCl (pH 7.5) 150 mM NaCl, 1 mM Na2EDTA, 1% Triton X-100) for 20 min. Cell extracts were incubated with primary antibodies for 2 h at 4 °C with gentle agitation. Protein G-Sepharose4 Fast Flow beads (GE Healthcare, Chicago, IL, USA) were added for an additional 1 h at 4 °C. Beads were gently washed three times in RIPA buffer. Bound proteins were eluted in 2× sample buffer and analyzed by western blot analysis.

### 4.10. Quantitative Reverse-Transcription Polymerase Chain Reaction (qRT-PCR)

Total RNA was extracted using GeneJet RNA Purification kit (Thermo Scientific), reverse-transcribed, and analyzed by quantitative PCR, as previously described [25]. The following primers were used: ADAM12 Fwd: 5-CAGGCACAAAGTGTGCAGAT-3, Rev: 5-GCTTGTGCTTCCTCCAAAGC-3, BSG Fwd: 5-GACGTCCTGGATGATGACGA-3, Rev: 5-GAAGAGTTCCTCTGGCGGAC-3, TGFβ1 Fwd: 5-CCGTGGAGGGGAAATTGAGG-3, Rev: 5-TGAACCCGTTGATGTCCACTT-3, TGFβ3 Fwd: 5-GTGCCGTGAACTGGCTTCT-3, Rev: 5-CAGCCCCAATCATCCACTCA-3, MXRA5 Fwd: 5-CCAGCCGAGAAAGACACAGT-3, MXRA5 Rev 5-AAGGTGGGCTTCGGGTATT-3, COL5A1 Fwd 5-GACTGCCAGATTTGGACACTAT-3, COL5A1 Rev 5-GGATGACCTTTACGAGGCTTAC-3, COL5A2 Fwd 5-CCAGGAGTTCCAGGTTTCAA-3, COL5A2 Rev 5-CAACTGTTCCTGGGTCACCT-3, FN Fwd 5-CCACAGTGGAGTATGTGGTTAG-3, FN Rev 5-CAGTCCTTTAGGGCGATCAAT’3. Ribosomal phosphoprotein (RPLP0) was used as a housekeeping reference gene, using primers Fwd: 5-CAGCAGTTTCTCCAGAGC-3, Rev: 5-TTCATTGTGGGAGCAGAC-3. Data were analyzed using the 2^−∆∆CT^ method.

### 4.11. In Vivo Experiments

Nude mice from Bomholtgård (Denmark) were used for growing MCF7-cell line. We used a previously characterized MCF-7 tet-off and tet-on model system, expressing ADAM12 [34].

### 4.12. Data Analysis

Raw data (CEL files) from the following datasets from different types of breast cancer were downloaded from Array Express (GSE2034, GSE4779, GSE6532, and GSE18229) and are available at Gene Expression Omnibus (GEO, http://www.ncbi.nlm.nih.gov/geo/, accessed on 23 February 2005). All data were normalized together using the Robust Multichip Average (RMA) approach, and calculations performed using the Subio platform program (https://www.subioplatform.com/products/subioplatform/, accessed on 1 March 2024). All calculations of data are on log2-transformed values. A simple linear regression analysis was performed where the expression level of ADAM12 (Affymetrix probe 213790_at) was correlated to the expression of COL5A1, COL5A2, and COL5A3. Pearson correlations were performed using GraphPad Prism.

### 4.13. Statistical Analysis

All experiments were performed at least three times. Student’s *t*-test was applied for comparing two groups and ANOVA for multiple group comparisons. The Kruskal–Wallis test was used to compare data in the box-and-whisker plot. *p* < 0.05 was considered statistically significant.

## Figures and Tables

**Figure 1 ijms-25-05871-f001:**
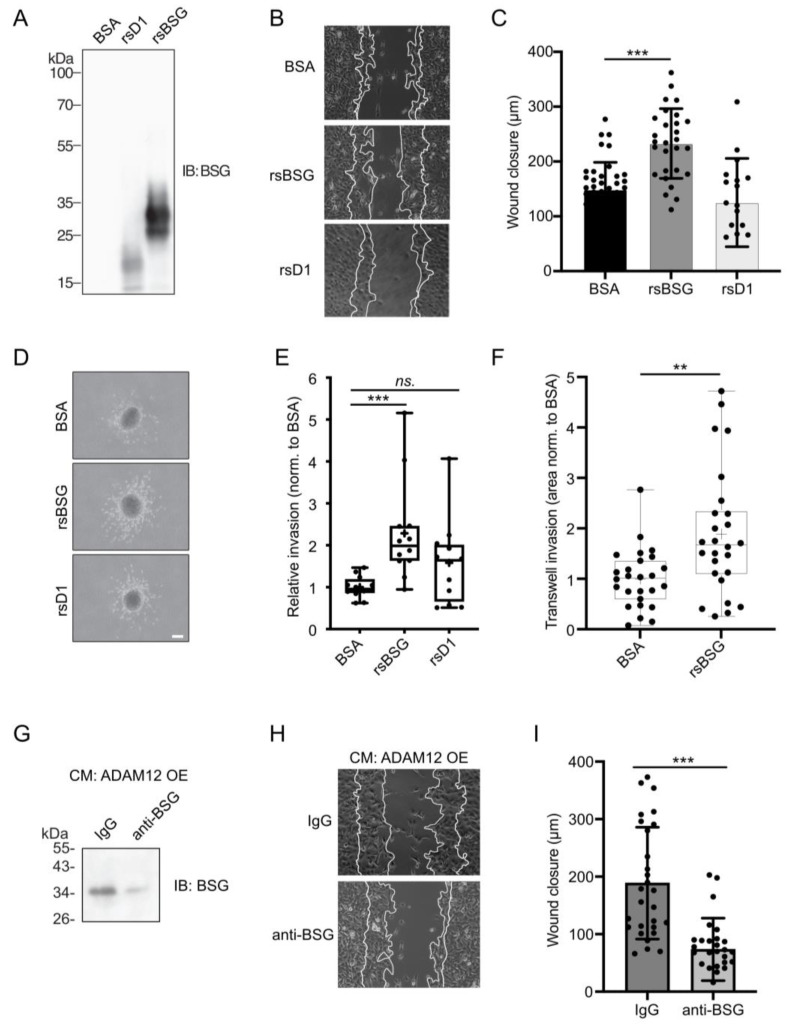
ADAM12-generated BSG ectodomain stimulates in vitro cancer cell migration and invasion. (**A**) Western blot analysis of two variants of recombinant soluble BSG (rsD1 and rsBSG) protein expressed in HEK293T cells and purified prior to analysis. rsBSG corresponds to ADAM12-generated soluble BSG and the truncated N-terminal fragment (rsD1) corresponds to the MMP14-generated soluble BSG. Forthcoming rsBSG will be used to describe the soluble fragment generated by ADAM12. (**B**) Representative images (scale bar = 100 μm) and (**C**) quantified distance of wound closure in HT1080 cell cultures treated for 6 h with soluble rsBSG, rsD1, or BSA control. (**D**) Representative images of MDA-MB-231 cell invasion in spheroids treated with 0.2 µg/mL rsBSG, rsD1, or BSA control for 24 h (scale bar = 100 μm) (**E**) Box-and-whisker plot showing the quantification of cell invasion in MDA-MB-231 spheroids seen in (**D**). Four independent experiments performed with triplicate technical repeats (median is shown as a line; mean is displayed as a cross). *** *p* < 0.001 using Kruskal–Wallis test. (**F**) Box-and-whisker plot showing the quantification of transwell invasion assay of MDA-MB-231 cells. Three independent experiments with individual measurements shown (media is shown as a line; mean is displayed as a cross). ** *p* < 0.003 using Mann–Whitney test. (**G**) Western blot analysis of soluble BSG in conditioned media from 293-VnR cells overexpressing ADAM12 (ADAM12 OE) with or without pre-clearing with anti-BSG antibody for 6 h. (**H**) Wound closure experiments of HT1080 treated with conditioned media from (**G**). (**I**) Quantified distance of wound closure in (**H**). Graphs in (**C**,**I**) represent means ± standard error of the mean (SEM) from at least 3 independent experiments. *** *p* < 0.005 using ANOVA and Student’s *t*-test, respectively.

**Figure 2 ijms-25-05871-f002:**
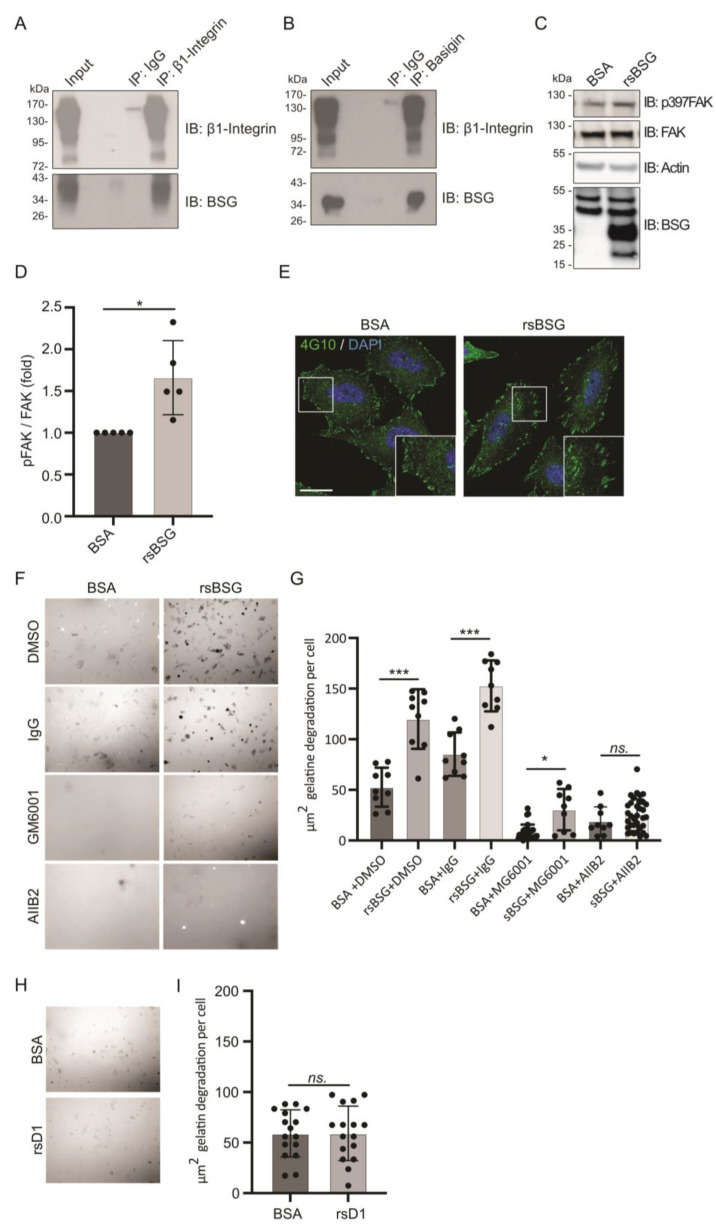
rsBSG binds β1 integrin and enhances integrin-dependent cell migration and gelatin degradation. (**A**,**B**) Co-immunoprecipitation of rsBSG and β1 integrin in HT1080 cells treated with rsBSG for 2 h before lysis. β1 integrin was immunoprecipitated using AIIB2 β1 integrin antibody and co-immunoprecipitated rsBSG was detected by western blot using an antibody against the strep-tag (**A**), whereas rsBSG was immunoprecipitated using the strep-tag antibody and co-immunoprecipitated β1 integrin was detected using the rat monoclonal AIIB2 anti-β1-integrin antibody (**B**). Total cell lysate (TCL) was tested for β1 integrin expression and the presence of rsBSG protein in the same blots. Control rat IgG immunoprecipitation served as negative control in both settings. (**C**) HT1080 cells were starved over night before being stimulated with 10 ug/mL BSA or rsBSG for 30 min. Samples were analyzed by western blot against p397FAK or total FAK. Actin served as an internal loading control. (**D**) Quantification of p397FAK relative to total FAK. (**E**) HeLa cells were starved overnight before being stimulated with 10 ug/mL BSA or rsBSG for 60 min. Cells were stained with anti-pTyr antibody (4G10), scale bar = 10 µm. (**F**) In situ gelatinase assay of HT1080 cells grown on Oregon green-labeled gelatin and treated with either BSA or rsBSG with either the AIIB2 antibody blocking β1 integrin activity or the metalloprotease inhibitor GM6001 for 6 h. Black areas represent gelatin degradation; scale bar = 50 µm. (**G**) Quantification of the area of gelatin degradation shown in (**F**), measured from 20 images using MetaMorph software. (**H**) In situ gelatinase assay performed as in F; cells were treated with 10 ug/mL BSA or BSG D1. (**I**) Quantification of gelatin degradation assay from (**H**), quantified as in (**F**). For all graphs, values represent means ± SEM from at least three independent experiments. * *p* < 0.05; *** *p* < 0.005, Student’s *t*-test or ANOVA.

**Figure 3 ijms-25-05871-f003:**
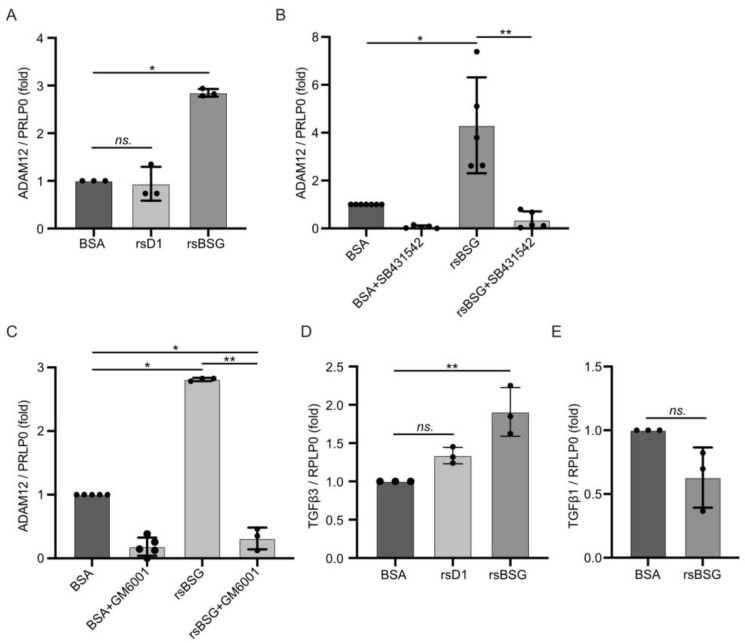
rsBSG stimulates ADAM12 and TGFβ3 expression, but has no effect on TGFβ1 expression. Quantitative RT-PCR showing fold mRNA expression of ADAM12 relative to the expression of RPLP0 in HT1080 cells treated for 24 h with control BSA, rsBSG, or rsD1 alone (**A**), together with the TGFβ1R inhibitor SB431542 (**B**), or with the metalloprotease inhibitor GM6001 as indicated (**C**). Quantitative RT-PCR showing fold mRNA expression of TGFβ3 relative to the expression of RPLP0 in HT1080 cells treated for 24 h with BSA, rsBSG or rsD1 (**D**), or TGFβ1 expression relative to RPLP0 in HT1080 cells treated with BSA or rsBSG for 24 h (**E**). All graphs represent means ± SEM from at least three independent experiments. * *p* < 0.05; ** *p* < 0.01, using ANOVA.

**Figure 4 ijms-25-05871-f004:**
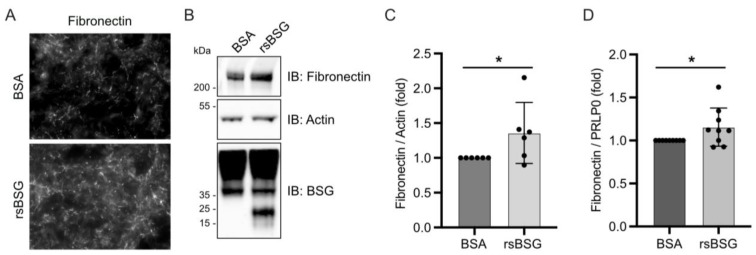
rsBSG stimulates fibronectin deposition. HT1080 WT cells were treated with 10 ug/mL rsBSG or BSA control for 24 h and analyzed for fibronectin deposition by immunofluorescence (**A**) and western blot analysis (**B**). (**C**) Densitometric quantification of fibronectin relative to actin from B; graph represents ± SEM from 3 independent experiments. Quantitative RT-PCR showing mRNA expression of fibronectin relative to PRLP0 (**D**). The graphs represent means ± SEM from 9 independent experiments. * *p* < 0.05 using Student’s *t*-test.

**Figure 5 ijms-25-05871-f005:**
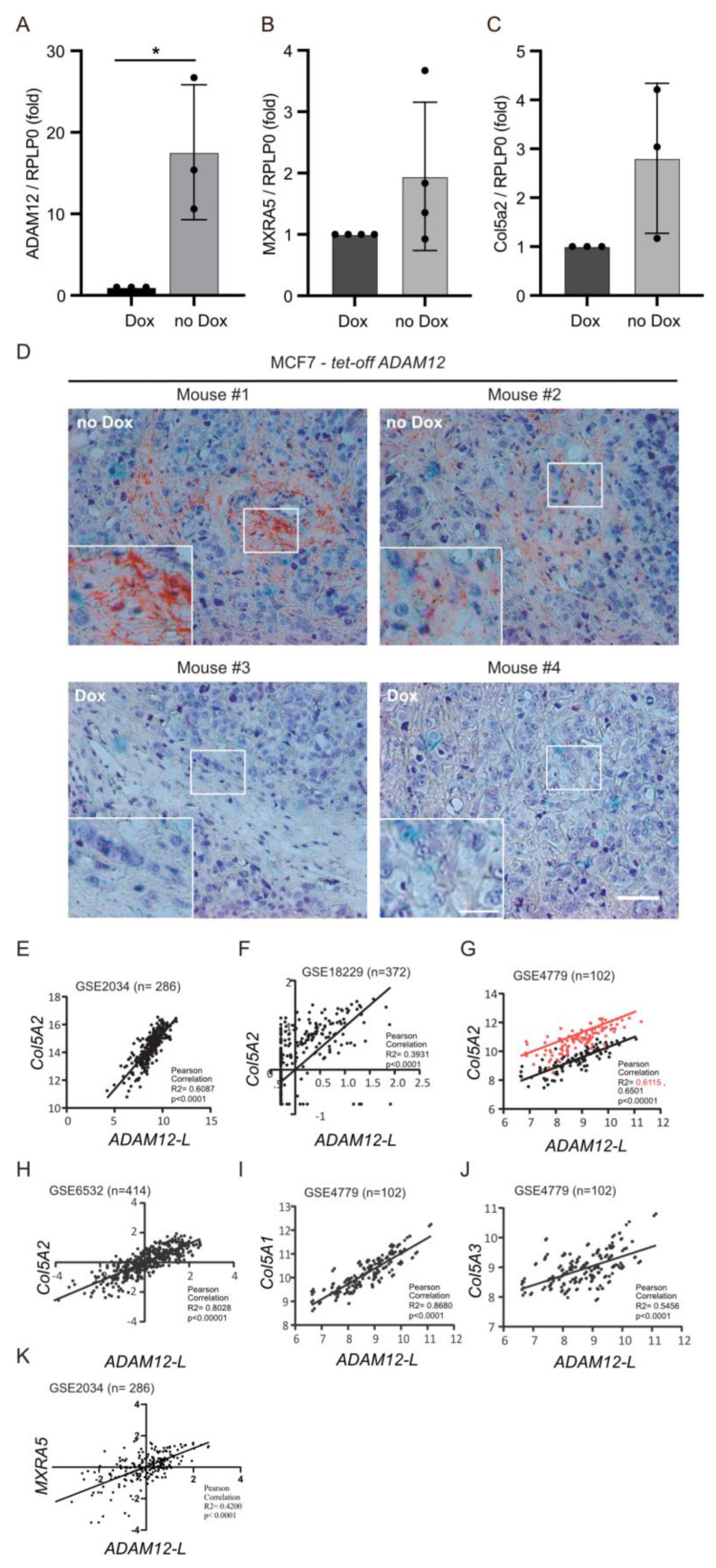
ADAM12 plays an important role in the synthesis of extracellular matrix proteins such as collagen type 5 in breast carcinomas. Quantitative RT-PCR showing fold mRNA expression of ADAM12 (**A**), MXRA5 (**B**), and Collagen5A2 (**C**) relative to the expression of RPLP0 in MCF7 cells expressing ADAM12 in a doxycycline (Dox)-dependent manner. All graphs represent means ± SEM from at least three independent experiments. * *p* < 0.05, using ANOVA. (**D**) Immunostaining of collagen type 5 together with Hematoxylin staining in MCF7 tumors in mice treated with or without Dox. Scale bar = 100 µm, insert = 20 µm. (**E**–**K**) Expression profile datasets from four different cohorts of human breast carcinomas. Positive correlation between mRNA expression of ADAM12 and expression of COL5A1, COL5A2, COL5A3, and MXRA5. The gene expression profile was assessed by a simple linear regression analysis and Pearson correlation.

## Data Availability

Public available GEO Array Expression datasets can be found as indicated in Data analysis (Section 4.12).

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
