# Peer review of "ADAM12-Generated Basigin Ectodomain Binds β1 Integrin and Enhances the Expression of Cancer-Related Extracellular Matrix Proteins"

_ijms, 2024, doi:10.3390/ijms25115871_

Round 1

Reviewer 1 Report

Comments and Suggestions for Authors

1.     For Figure 1B and 1G, please enlarge the images for better visualization of wound closure.

2.     For Figure 2E, some images such as the group of rsBSG+GM6001 are out of focus, please provide images with higher quality.

3.     The in vivo assessment of ADAM12 and collagen type 5 in the study was weak, authors only conducted immunostaining of tumor sections. Including tumor challenging and survival results will increase the impact of the work.

Author Response

We have uploaded a full reply as a PDF for the easy of reading and we have included a figure as well.

Reviewer 2 Report

Comments and Suggestions for Authors

In the current work the authors aim at exploring the potential role of basigin ectodomain in cancer. They describe that the interaction of basigin ectodomain with b1-integrin modulates the expression of tumor-associated ECM proteins and promote migration. This study follows up on previous studies from the authors.

Specific comments:

-        Figure 1 – In the legend “Graphs in (C+H) represent means ± standard error of the mean (SEM) from at least 3 independent experiments. ***p<0.005 using Students T-test or ANOVA, respectively.” C is ANOVA and H student T-test and not the other way around. This should be corrected.

-        “For comparison, we tested whether the rsD1 fragment was able to induce gelatine degradation, but found no significant changes com-pared to the BSA control (not shown) .”  - in light of the data presented on Figure 1, this data will add extra value in the manuscript and should be incorporated

-        “Further, as a positive feedback loop between BSG and TGFβ has been reported [32],” – sentence needs to be rephrased

-        “Western blot analysis indicated a moderate increase in fibronection protein levels when cells were treated with rsBSG (Fig. 4B), and” – did the authors quantified the WB and there were no differences? They should comment on how these results and the immunofluorescent can co-exist. Immunofluorescence is not a rigorous quantitative method and has many variables. Also note that “fibronectin” is misspelled in the sentence. Authors should highly consider looking into COL5A2 mRNA expression in this model.

-        “It is well exstablished that human breast carcinomas exhibit increased deposits of extracellular matrix” – “established” is missplelled.

-        “Specifically, we used a previously characterized MCF-7 tet-off model system [34].” – authors should specify in this sentence which gene is under the control of the tet-off system.

-        Correlation with the human datasets should be extended to other components that were identified as important including TGFB3, MXRA5

Comments on the Quality of English Language

English requires careful revision

Author Response

We have uploaded a full reply as a PDF for the easy of reading

Reviewer 3 Report

Comments and Suggestions for Authors

This paper describes how soluble BSG derived from ADAM12 signaling can stimulate cell responses in tumoral cell migration and invasion. The methods are rigorous and the results are clearly shown in the manuscript. I strongly suggest the authors increase the size and improve the distribution of images to display the results better - for instance, the gelatinase assay is too small, and degradation spots are not indicated clearly in the BSA+DMSO or BSA+IgG groups, where they are more subtle. Still in this assay, if cells were used to normalize the area degraded by cells, authors should display them in images as well, if possible.

In Figure 3, authors should display the data points instead of using a dynamite plot. 

Why did the authors opt for the MCF7 cell line instead of a murine BC strain (e.g., 4T1)?

Regarding text structure, the last sentence of the Introduction section needs to be revised. There's a typo on the first line of topic 2.5, "exstablished". Double-spacing was found in the Abstract, Results and Discussion sections.

Author Response

We have uploaded our response as a PDF

Reviewer 4 Report

Comments and Suggestions for Authors

I reviewed the article "ADAM12-Generated Basigin Ectodomain Binds β1 Integrin and Enhances the Expression of Cancer-Related Extracellular Matrix Proteins", presented by Mygind et al,. The study try to elucidate the role of basigin ectodomain in cancer progression. Here are some observations that could complement this research and provide deeper insights:

While the study demonstrates binding between basigin ectodomain and β1 integrin, further investigation into integrin activationcould be valuable. Assays like ligand binding assays or IPP downstream signaling pathway activation (in conjuction with BSG-b1 integrin presneted results or FAK phosphorylation) could confirm β1 integrin's functional role in the observed effects.

While the study shows increased migration, a transwell invasion assay could be used to assess the ability of basigin to promote cancer cell invasion through a basement membrane model, mimicking the initial step of metastasis.. what about the proteolytic activity by zymograms?

Implementing a metastasis model would strengthen the link between basigin ectodomain and cancer progression. Analyzing metastasis rates and organ colonization by cancer cells could solidify the findings to complement .

Further exploration of the specific pathways downstream of β1 integrin activation and their interaction with TGFβ3 signaling could elucidate the mechanism by which basigin ectodomain contributes to desmoplasia.

blocking the TGFβ3 signaling pathway can abrogate the basigin-induced changes in ECM protein expression? Please discuss

The study uses ADAM12-expressing cells. What would be expected with cells with silenced or knocked-out ADAM12 expression? These assays could provide a clearer picture of how ADAM12 activity directly influences basigin shedding and downstream effects.

Author Response

(The authors gave the same response as above.)

Round 2

Reviewer 4 Report

Comments and Suggestions for Authors

The authors have appropriately addressed the questions raised and have improved the article.